# Identification and characterization of epicuticular proteins of nematodes sharing motifs with cuticular proteins of arthropods

**Bruno Betschart**[1][◉]*, **Marco Bisoffi**[2][◉], **Ferial Alaeddine**[1][◉]

1 Institute of Biology, University of Neuchâtel, Neuchâtel, Switzerland, 2 Chemistry and Biochemistry, Schmid College of Science and Technology, Chapman University, Orange, California, United States of America

◉ These authors contributed equally to this work.
* bruno.betschart@unine.ch

**Data Availability Statement:** In addition to the data in the Supporting information file, relevant data have been submitted to Dryad (DOI: 10.5061/dryad.fttdz08vs). Nucleotide sequence data reported are available in the Third Party Annotation

## Abstract

Specific collagens and insoluble proteins called cuticlins are major constituents of the nematode cuticles. The epicuticle, which forms the outermost electron-dense layer of the cuticle, is composed of another category of insoluble proteins called epicuticlins. It is distinct from the insoluble cuticlins localized in the cortical layer and the fibrous ribbon underneath lateral alae. Our objective was to identify and characterize genes and their encoded proteins forming the epicuticle. The combination between previously obtained laboratory results and recently made available data through the whole-genome shotgun contigs (WGS) and the transcriptome Shotgun Assembly (TSA) sequencing projects of *Ascaris suum* allowed us to identify the first epicuticlin gene, *Asu-epic-1*, on the chromosome VI. This gene is formed of exon1 (55 bp) and exon2 (1067 bp), separated by an intron of 1593 bp. Exon 2 is formed of tandem repeats (TR) whose number varies in different cDNA and genomic clones of *Asu-epic-1*. These variations could be due to slippage of the polymerases during DNA replication and RNA transcription leading to insertions and deletions (Indels). The deduced protein, Asu-EPIC-1, consists of a signal peptide of 20 amino acids followed by 353 amino acids composed of seven TR of 49 or 51 amino acids each. Three highly conserved tyrosine motifs characterize each repeat. The GYR motif is the Pfam motif PF02756 present in several cuticular proteins of arthropods. Asu-EPIC-1 is an intrinsically disordered protein (IDP) containing seven predicted molecular recognition features (MoRFs). This type of protein undergoes a disorder-to-order transition upon binding protein partners. Three epicuticular sequences have been identified in *A. suum*, *Ascaris lumbricoides*, and *Toxocara canis*. Homologous epicuticular proteins were identified in over 50 other nematode species. The potential of this new category of proteins in forming the nematode cuticle through covalent interactions with other cuticular components, particularly with collagens, is discussed. Their localization in the outermost layer of the nematode body and their unique structure render them crucial candidates for biochemical and molecular interaction studies and targets for new biotechnological and biomedical applications.

Section of the DDBJ/ENA/GenBank databases (accession numbers TPA: BK061351 and TPA: BK061342).

**Funding:** The work was founded by following grants: BB 008439 Swiss National Science Foundation (SNSF) BB 26764 Swiss National Science Foundation (SNSF) BB 33961 Swiss National Science Foundation (SNSF) BB 047194 Swiss National Science Foundation (SNSF) The funders had no role in study design, data collection and analysis, decision to publish, or preparation of the manuscript.

**Competing interests:** The authors have declared that no competing interests exist.

## Introduction

The phylum Nematoda, classified with Arthropoda in the clade Ecdysozoa, inhabits a broad range of environments. Nematodes are present in freshwater, marine, and diverse terrestrial environments accounting for about 80% of all animals [1]. Free-living species play an essential role in the decomposition of organic material. Other nematodes parasitize humans, vertebrate animals, or insects. Finally, nematodes can also be parasites of plants, causing significant crop losses [2]. Nematodes undergo up to four molts during their life cycle, thereby shedding the old cuticle and replacing it with a new one [3, 4]. The nematode cuticle is a complex extracellular structure whose morphology varies considerably between species and developmental stages [5]. The cuticle, acting as a protective exoskeleton, can be associated with glycoproteins and lipids [6–10]. Several glycoproteins are associated with a surface coat and are secretory products involved in the interaction of the nematodes with their environment [11, 12]. Other glycoproteins, for example, filariae's 29 kDa glycoprotein (gp29), a homolog of the enzyme glutathione peroxidase (GSHPx), is considered intrinsic to the cuticle [13, 14]. Intrinsic proteins are synthesized in the hypodermic layer and secreted into the cuticle [9]. Gp29 could be involved in the defense against immune-mediated cytotoxicity or in catalyzing the formation of crosslinking residues, such as dityrosine, trityrosine, and isotrityrosine [13]. Such residues are present in cuticular collagens and insoluble proteins and shape the cuticle through inter- and intra-molecular crosslinks [10, 15, 16]. Genes coding for insoluble cuticular proteins were identified in several nematodes, i.e., *C. elegans* [17, 18], *Meloidogyne artiellia* [19], *A. lumbricoides* [20], *Brugia pahangi*, and *Brugia malayi* [21]. These genes were called *cut-1* to *cut-6*. *Cut-1* encodes a secreted protein of 423 amino acids with a zona pellucida (ZP) domain and is expressed in the dauer larva of *C. elegans* in a ribbon lying beneath the alae [22]. In filariae, CUT-1 is restricted to the median layer of the cuticles [21]. ZP domains are also identified in cuticlins encoded by *cut-3* to *cut-6* [23]. In contrast, *cut-2* of *C. elegans* has no ZP domain and is transcribed in all stages before molting [18]. The CUT-2 protein, composed of 215 amino acids, has been immunolocalized in the cortical layers of the cuticle, but not in the electron-dense epicuticle [22]. The protein contains several short repeats, also found in structural proteins of the cuticles or eggshells of various insects [18]. All insoluble cuticular proteins are crosslinked via di- and tri-tyrosines [16, 18, 24–26].

The electron-dense layer at the exterior boundary of nematode cuticles is defined as the epicuticle [27]. Using *A. suum*, Fujimoto and Kanaya [24] manually separated the cuticles from the body carcass. Cuticular collagens were removed through digestion with bacterial collagenase. They termed the remaining insoluble material cuticlin, having high contents of proline and alanine. Electron microscopy of the isolated material showed the presence of electron-dense structures [24], typical for the epicuticle. We showed that polyclonal antibodies prepared against the isolated insoluble material of *A. suum* reacted with the cortical outermost epicuticular layers of *A. suum* and filariae [25, 28]. Immunoscreening of an *A. suum* cDNA library with a monoclonal antibody specific for the insoluble fraction allowed the isolation of a cDNA clone with an incomplete coding region of 644 nucleotides (NCBI GenBank accession number X92101.1) [29]. The deduced amino acid sequence had six repetitive peptide motifs containing three tyrosine residues each. As described by Fujimoto for his cuticlin, the protein is rich in proline and alanine residues [24, 29]. A fusion protein was produced to raise specific antibodies, and the epitope's presence within the epicuticular layer of *A. suum* was confirmed [29]. Additional cDNA and genomic clones were isolated, showing the typical nucleotide repeat patterns. The partial gene was named *Asu-epicut1* [30] underlining that its encoded protein is in the epicuticle. The early data suggested that the 5' and 3' parts of the cDNA clones would

belong to two different exons separated by an intron in the gene. No full-length gene structure could be inferred before genomic sequencing.

Significant technological advances recently allowed the complete genome of *A. suum* to be sequenced [31]. We reverified, completed, and identified the *Asu-epicut1* sequences using this new database. Using *Asu-epicut1* as a query, we identified two additional epicuticular genes, *Asu-epic-2* and *Asu-epic-3*. Homologous proteins are found in over 50 nematode species of three different nematode clades. The possible role and mode of interaction of epicuticlins with cuticular collagens are discussed based on their typical intrinsically disordered protein characteristics, tyrosine motifs, and cysteine residues. Finally, the relationship of this new category of epicuticular proteins with proteins present in Arthropoda is elucidated.

## Material and methods

The cDNA (X92101.2, 31B1A; AJ408885.1, C1; AJ408886.1, C2; AJ408887.1, C3) and genomic clones (AJ408888.1, G1; AJ408889.1, G2; AJ408890.1, G3), coding for *A. suum* epicuticlins, were compared through alignments using Jalview [32]. Based on their conserved nucleotide sequence (aagaggaa) (Fig 2), Blasts were carried out on the following databases: NCBI (https://blast.ncbi.nlm.nih.gov/Blast.cgi), Nucleotide collection and whole-genome shotgun contigs (wgs) of *A. suum*; WormbaseParasite [33] and the EMBL-EBI services [34]. The *A. suum* TSA sequences JI177333, JI176387, JI178090, and JI180250, were used in their reversed complementary version (rc). All former *A. suum* epicuticlin sequences were called "epicut". In agreement with the nomenclature directives of Wormbase the new epicuticlin genes are termed "epic". The complete *Asu-epic-1* gene was identified through blasts using the 5' part of 55 nucleotides of cDNA/TSA clones starting with ATG and the 3' part formed of the repeat-containing region (S1 and S2 Figs). The *Asu-epic-1* and *Asu-epic-2* nucleotide sequence data reported are available in the Third-Party Annotation Section of the DDBJ/ENA/GenBank databases under the accession number TPA: BK061342 and TPA: BK061351. Additional data are deposited in the Dryad, Dataset, https://doi.org/10.5061/dryad.fttdz08vs.

Protein Blasts were carried out mainly on the protein knowledge platform UniProt [35] and the same platforms as the nucleotide sequence Blast. Multiple or pairwise nucleotide or protein sequence alignments were carried out using the EMBL-EBI services: Clustal Omega for multiple alignments and Needle or Matcher for pairwise alignments. Radar was used to automatically align protein repeats and Phobius to identify signal peptides [34]. The Multiple sequence viewer [36] was used to compare nucleotide and protein repeat sequences. Wormbase was the authoritative data source for the *C. elegans* genes [37]. Expasy services (https://www.expasy.org/) [38] were applied for the in-depth study of the epicuticlins: STRING [39] and Compute pI/MW [40]. The presence of intrinsically disordered regions in the proteins was calculated using the IUPred2A tool (https://iupred2a.elte.hu/) [41], and the characteristics of the IDPs were analyzed using CIDER [42]. For the prediction of MoRFs, we used the software tool MoRFchibi | Gsponer Lab (ubc.ca) [43]. The *C. elegans* collagen database, CeColDB (http://CeColDB.permalink.cc/), was used to analyze the structure of cuticular collagens.

## Results

### Identification of the first *A. suum* epicuticular gene, *Asu-epic-1*

Blast with the *A. suum* cDNA AJ408887 resulted in thirty nucleotide matches with > 95% nucleotide identity to the query, comprising cDNA sequences (including our previously isolated cDNA clones: X92101, AJ408885, AJ408886, AJ408887, Fig 1), TSA sequences, and Expressed Sequence Tag (EST) sequences (S1 Table).

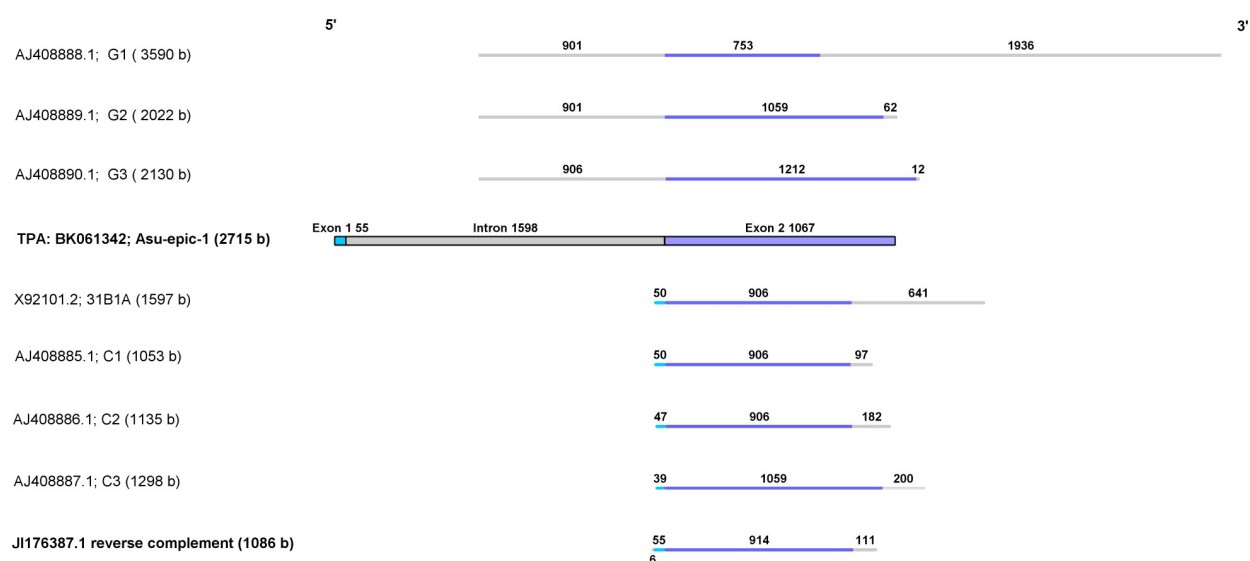

**Fig 1. Sequence comparison of different *Asu-epic-1* clones.** Formerly characterized genomic (G) and cDNA clones (C) are aligned with the TSA sequence JI176387.1rc and the complete *Asu-epic-1* gene of *A. suum*. Exons 1 and 2 are indicated as light and darker blue boxes or lines. The intron and untranslated regions are indicated as a gray box and gray lines with the number of nucleotides, respectively. The GenBank accession numbers, the initial designation used for the sequence submission [29, 30], and the sequence length (bases, b) are given.

Only six entries had a 5' start codon (S1 Fig). Two TSA sequences (JI176387rc and J178090rc) have the ATG initiation codon, an unrepeated sequence stretch, followed by six repeats. While JI176387rc ends with the termination codon, the J178090rc sequence does not. Otherwise, the two are 100% identical. The other four EST fragments (BI781775, BI781825, BI782671, and BI781932) also contain the correct initiation codon. Complementary to the cDNA/TSA sequences, thirteen different genomic sequences with more than 95% nucleotide identities to the query were identified in three genome databases. Three were, as expected, the genomic clones already described (Fig 1). The most recent genomic sequence was from the *A. suum* isolate RED_2019 chromosome 6, whole genome shotgun sequencing project (PRJNA62057) [31] with a 99.2% nucleotide identity to the query. The sequence JI176387rc was considered the appropriate mRNA (Fig 1 and S1 Fig) from which to build a gene model (S2 Fig). The proposed gene is localized on the negative strand of chromosome VI and comprises two exons, separated by an intron of 1593 nucleotides (Fig 1 and S2 Fig). The *Asu-epic-1* gene starts with a signal sequence of 60 nucleotides followed by seven TRs varying in length between 147 and 153 nucleotides, and the last repeat ends with a stop codon (Fig 2).

The seven repeats were aligned, and specific nucleotide differences were detected (Fig 2). For example, an adenine replaces a thymidine in position 120 of repeat five. These patterns helped manually optimize the automatically generated alignments of one TSA and four cDNA sequences. They were individually compared with the identified *Asu-epic-1* gene (Fig 3), and the presence of the specific nucleotide patterns in the diverse clones was confirmed.

TR1 and TR2 of all sequences are identical to the gene except for one difference in four cDNA sequences (substituting GC by CG in positions 61–62 within TR 1). All sequences end with a termination codon, indicating that they are complete and that their repeat numbers (six-seven) are defined. AJ408887 is the only cDNA sequence containing seven repeats as *Asu-epic-1*. However, the TR4 is duplicated while TR5 is missing. In repeats three to five, all sequences showed variations produced by Indels.

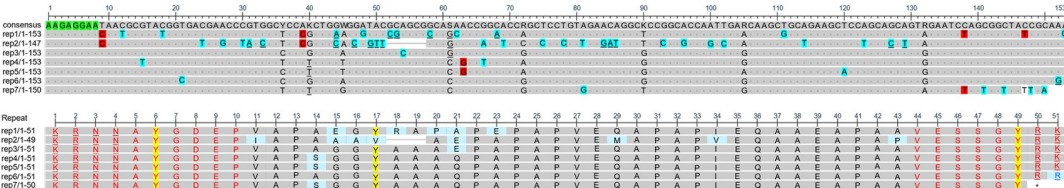

**Fig 2. Characteristics of the aligned nucleotide and deduced amino acid repeats of the gene *Asu-epic-1*.** A gap is introduced in repeat 2 for an optimal alignment. The seven aligned repeats (without the signal peptide sequence) vary in length from 147 to 153 nucleotides and start with a conserved motif of eight nucleotides (green). The seven repeats are individually colored. These colors are reused in Fig 3. Nucleotide variations between repeats are highlighted in blue, red, or black letters. Blue highlighted nucleotides are specific for a given repeat. Nucleotides highlighted in red are present in two repeats. Black letters of nucleotides show at a given position variations in more than two repeats. Underlined nucleotides lead to changes in the amino acid residues. In the amino acid repeats, the regions at the N- and C-terminal sides are conserved (red letters). Underlined red amino acids represent hydrophilic motifs. Each of the conserved region contains one tyrosine residue (highlighted in yellow). A third tyrosine (yellow) is present in between, except in repeat 2. Amino acid variations between the repeats are highlighted in blue. Repeat 6 ends with an arginine (blue) instead of the usual lysine.

## Properties of the deduced Asu-EPIC-1 protein

The deduced protein of Asu-EPIC-1 has 373 amino acids (37.64 kDa) and an estimated iso-electric point of 4.28 (calculated with Compute pI/MW). It starts with a signal peptide of 20 amino acids (S3 Fig). Asu-EPIC-1 has seven TRs with a length of 51 (repeats 1 and 3–6) and 49 amino acids (repeats 2 and 7). The aligned amino acid sequences of the repeats showed high conservation of the N- (KRNNA**Y**GDEP) and C-terminal (VESSG**Y**RK) parts, including two of the three tyrosine residues. Some amino acid variations are present between these two con-served parts, mainly in TR1 and TR2 (Fig 2). Repeat 2 lacks two amino acids, has only two tyrosine residues, and contains more variations (8) than the other repeats. Repeat 7 ends with the third tyrosine (Fig 2). Importantly, the different deletions and insertions found in several clones (Fig 3) did not change the reading frame and did not result in amino acid variations (S3

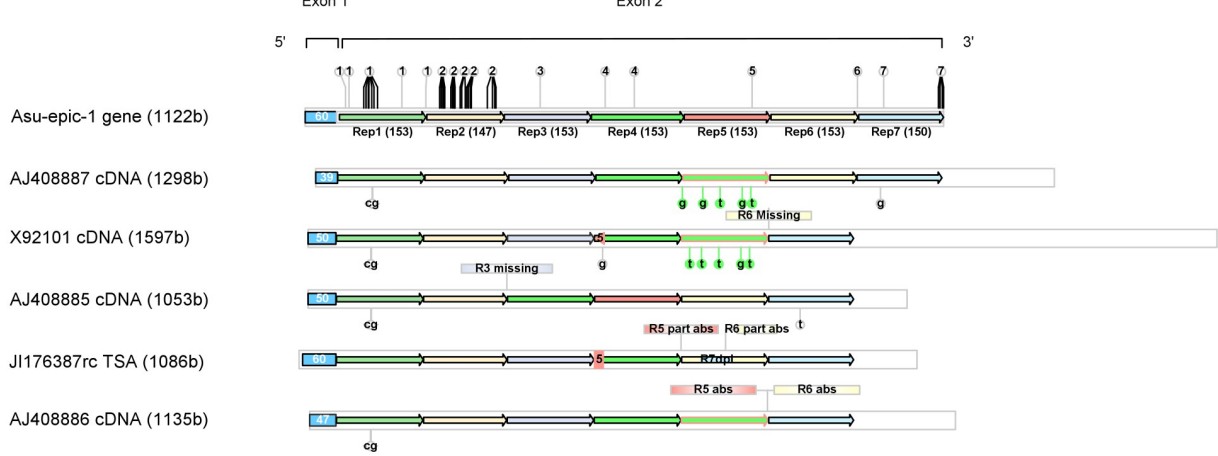

**Fig 3. Schematic representation of the nucleotide sequence of the *Asu-epic-1* gene and five cDNA/TSA sequences.** The nucleotide sequences (total length in parenthesis) are represented by light gray bars. The exon1 coding for the signal peptide is indicated as a light blue box. Arrows in different colors represent the seven tandem repeats of exon two (see Fig 2). The specific nucleotides of a given repeat of the *Asu-epic-1* sequence are shown as numbers above their position in the repeat sequence. In the cDNA/TSA sequences, missing repeats are indicated above their correct corresponding position. Duplicated repeats or parts of them are represented with the color of the corresponding repeat. Their colored borders indicate the type of missing repeat. Single nucleotide differences in comparison to the *Asu-epic-1* gene are indicated below the sequences.

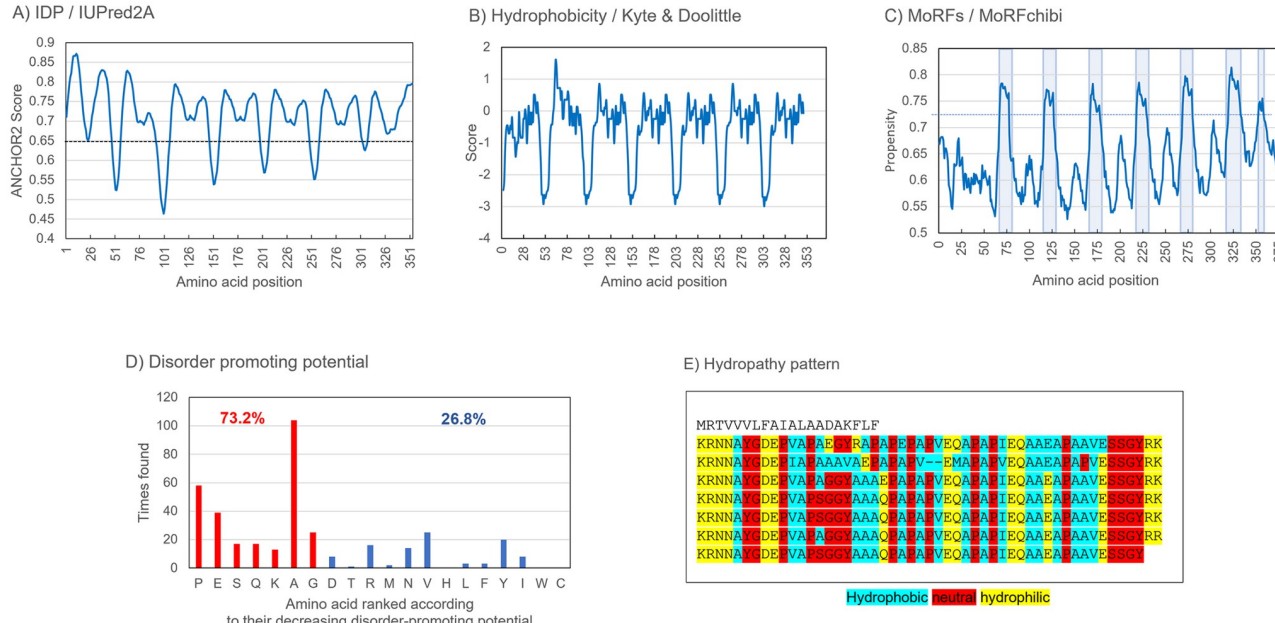

**Fig 4. Physicochemical properties of Asu-EPIC-1.** A) The protein has a highly intrinsically disordered profile as predicted by IUPred2A [41]. The ANCHOR2 score of 0.65 is a threshold indicative of intrinsically disordered regions. B) Hydrophobicity plot, according to Kyte & Doolittle, shows six hydrophilic regions. C) Identification of seven putative molecular recognition features (MoRFs) (blue regions) using the MoRFchibi SYSTEM [43]. The cut-off level is set at 0.72 (blue line). The first six MoRFs are the parts of the conserved amino acid regions, and each contains two of the three tyrosine residues (SG**Y**RKKRNNA**Y**). The last MoRF is associated with the sequence QAPAPI. D) The distribution and proportion of disorder-promoting amino acids (in red) versus more order-promoting amino acids (blue) [44]. E) Distribution of the hydropathy pattern in the TRs according to Pommié et al. [45].

Fig). Only one substitution (arginine instead of lysine) at the end of repeat six was found in JI176387rc, and the nucleotide differences in repeat one of the four cDNA sequences replaced the alanine with arginine at position 41 of the first protein repeat (R instead of A) (S3 Fig).

The deduced Asu-EPIC-1 protein is an intrinsically disordered protein (IDP), as shown by computational analysis (Fig 4A), with 73.2% of the amino acids having disorder-promoting characteristics (Fig 4D).

The hydrophobicity plot indicates six hydrophilic regions formed each by the four amino acids RKKR or RRKR with scores of up to -2.989 (Fig 4B). The hydropathy pattern confirms the hydrophobicity profile with six blocks of hydrophilic amino acid stretches of RKKRNN (Fig 4E). In the first repeat, the four hydrophilic amino acids KRNN confer less hydrophilic property than the six hydrophilic amino acids of the other repeats. Since Asu-EPIC-1 is predicted as an IDP, it would not fold into a defined or fixed three-dimensional structure but changes its form according to potential interacting partners. Seven molecular recognition features (MoRFs) are effectively detectable in Asu-EPIC-1 (Fig 4C). Six correspond to the conserved regions, including two tyrosine residues. The last questionable MoRF starting at position 353 (QAPAPI) differs from the others and does not include tyrosine. The presence of three tyrosine residues per TR (except TR2 with two tyrosine residues) and two in each MoRF points to their crucial role in the assembly of the epicuticlin structure. The conserved tyrosine motif GYR at the end of each repeat is an annotated Pfam motif (PF02756) found in various cuticular IDPs of insects. The protein B4IZ60 of *Drosophila grimshawi* was selected to compare the architecture of these motifs with the tyrosine motifs of Asu-EPIC-1 (Fig 5).

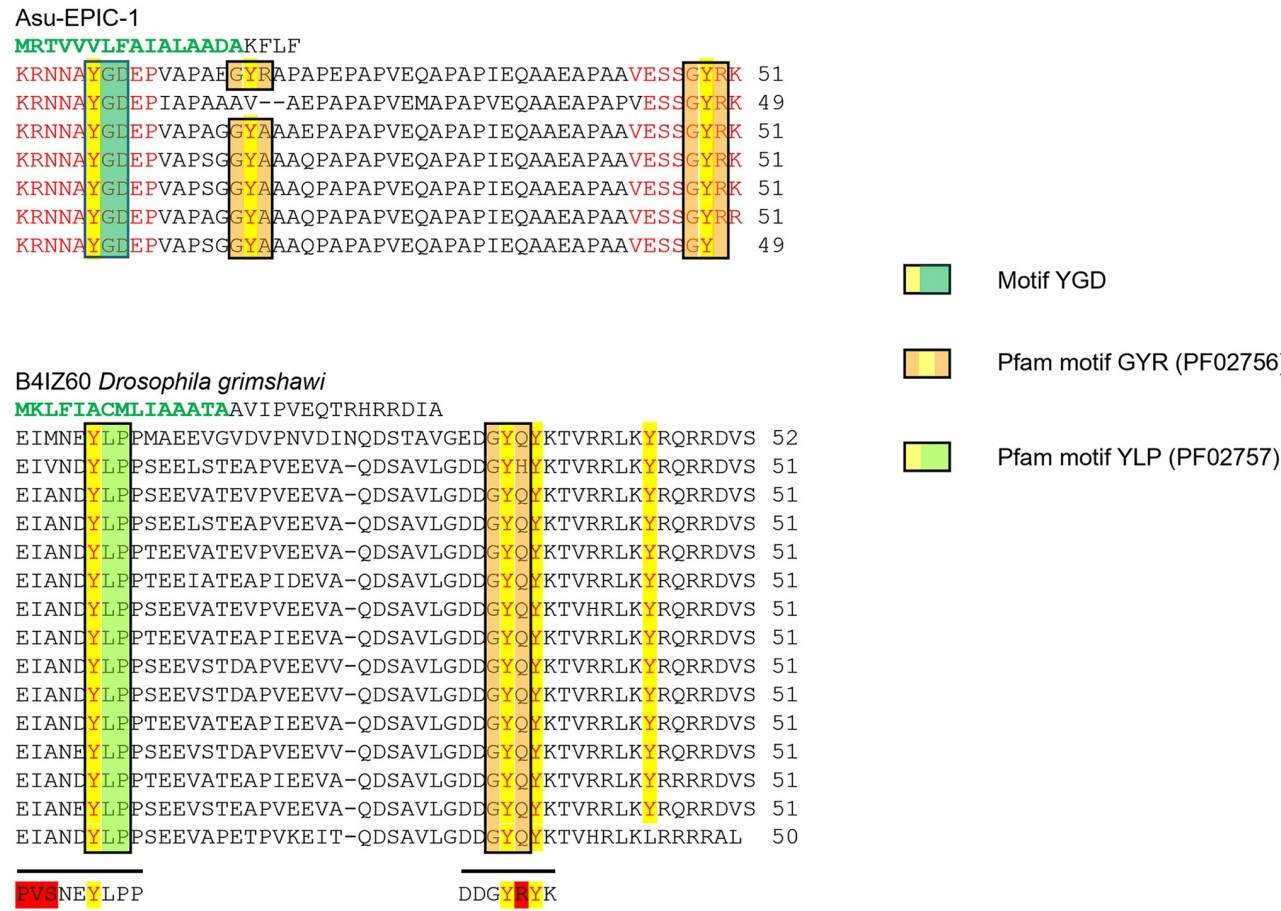

**Fig 5. Comparison of Pfam patterns in Asu-EPIC-1 and the protein B4IZ60 of *D. grimshawi*.** The signal peptide sequence is in green letters. The *Drosophila* protein is organized analogically to the repeat units of Asu-EPIC-1. The number of amino acid sequences per TR is indicated on the right side. Gaps were introduced for better alignments. The red letters in Asu-EPIC-1 represent the highly conserved regions located at each repeat's N- and C-terminal ends. All tyrosine residues (in red letters) are highlighted in yellow. The tyrosine motifs are boxed (orange = Pfam02756; light green = Pfam02757 and green = motif YGD). The horizontal lines below the *D. grimshawi* repeat sequences represent the amino acids most frequently present in all *Drosophila* matches on the Interpro website (http://www.ebi.ac.uk/interpro/) [46]. The red background letters indicate the amino acids that differ from the *D. grimshawi* sequence.

The repeats of both species have an almost identical length of 51 amino acids, and both proteins are IDPs and have a signal peptide. The fifteen TRs of the insect protein have four tyrosine residues per repeat. Three are organized in the two Pfam annotated motifs YLP and GYR [40]. In both proteins, the motifs YGD and YLP are at the same N-terminal position of each repeat. The following tyrosine residue is in the GYR motif or a variant (GYA, resp GYQY). The last tyrosine residue is present in the C-terminal region of the repeats within the GYR motif in Asu-EPIC-1 and within the not annotated KYR motif in *D. grimshawi* repeats.

## Dentification of additional nematode epicuticlins

Blasts with parts of the Asu-EPIC-1 sequence as a query on the SIB-UniProtKB and the NCBI protein databases rendered over one hundred unidentified epicuticular protein sequences belonging to 54 different nematode species of the class Chromadoria (Spirurina, clade III; Tylenchina, clade IV; Rhabditina, clade V) (S2 Table). The number of repeats varies from one (in several species) to maximally 33 repeats in *T. canis* (A0A0B2V0B7). Common to all

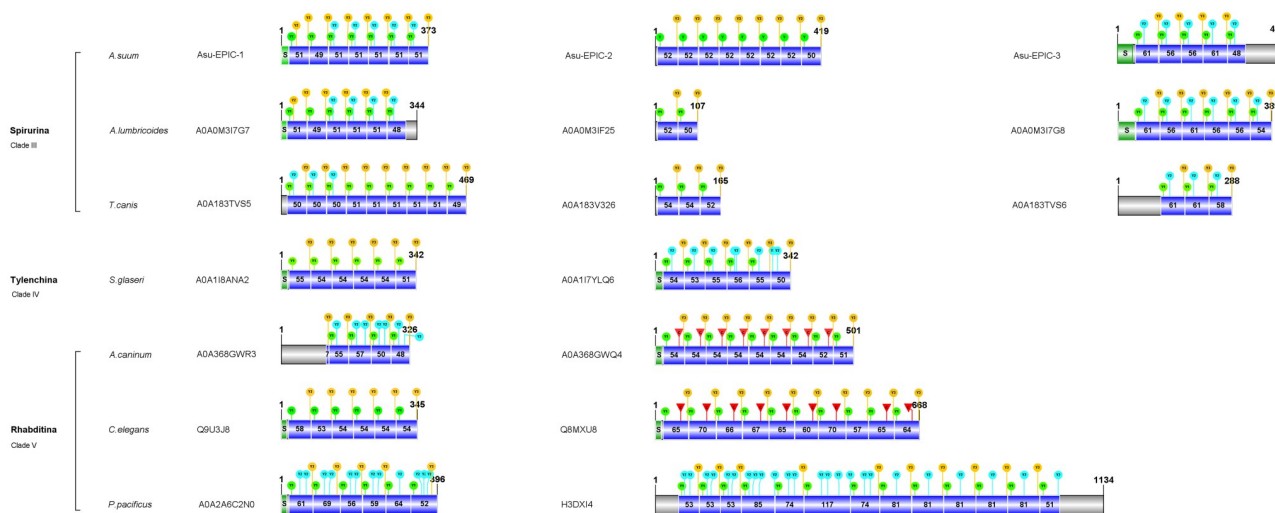

**Fig 6. Comparison of the epicuticlins of seven nematode species belonging to three different clades.** The sequences, with the Uniprot accession numbers, are drawn to scale, based on the number of amino acids, using IBS version 1.0.3 [47]. The different epicuticlin sequences of a given species are horizontally positioned. Signal peptide sequences (S) are shown as light green boxes. Repeats are represented as blue boxes in which the number of amino acids is given. Grey regions are non-repeat stretches. For each repeat, tyrosine residues are indicated with symbols (light green circles Y1 = **Y**GD; yellow circles Y3 = G**Y**R; the blue circles = Y2 indicate the additional tyrosine). Cysteine residues (red triangles C) are present in several epicuticlins of nematode species belonging to Clade V. Additional information is available in the S2 Table and deposited in the Dryad, Dataset, https://doi.org/10.5061/dryad.fttdz08vs).

sequences is the presence of the two conserved epicuticlin amino acid motifs GYR and YGD (Fig 6).

All these putative epicuticlin proteins are IDP and have alanine and proline-rich patterns. Some, but not all of them, have a signal peptide. The number and structure of TRs vary within a species, between species, and between the three clades. The repeats' length ranges from 49–71 amino acids. Several nematode species have two to three distinct epicuticlin proteins (Fig 6 and S2 Table), which differ in their amino acid sequences, the number, and length of the repeats, and the number of tyrosines (two to four) per repeat. Many species of clade V have two different epicuticlins, one of which contains a cysteine localized between the conserved regions (Fig 6 and S2 Table). No cysteine was found in the epicuticular repeats of the other nematodes. Besides EPIC-1, two different epicuticlins, named EPIC-2 and EPIC-3, were identified in the databanks of *A. suum*, *A. lumbricoides*, and *T. canis* (see below for more details). An appropriate and consistent overall classification of the epicuticlins is at present impossible. The epicuticular proteins of seven nematode species are presented in more detail to illustrate their differences within the clades III to V. The widely used model nematode *C. elegans* is included in clade V.

## Clade III Spirurina: *A. suum*, *A. lumbricoides* and *T. canis*

Asu-EPIC-2 protein is encoded by the gene *Asu-epic-2*, located on chromosome III (isolate RED_2019) in the position 10824598–10825857 of the negative strand. The deduced protein has 419 residues (42.79 kDa) with an estimated isoelectric point of pH 4.10. It consists of an NH2-terminal unrepeated stretch of five amino acids followed by eight TRs. The seven first repeats consist of 52 amino acids and the last one of 50 amino acids. Each repeat contains only two tyrosine residues (Fig 6). The highly conserved tyrosine regions differ from Asu-EPIC-1 by five amino acids (underlined) KRNNQ**YGD**EP and VQAS**GYR**R. The third *A. suum*

epicuticlin sequence was identified in a genomic DNA fragment annotated AgB12_g028 (on scaffold AgB12:654,298–657,498 reverse strand). This sequence, named *Asu-epic-3*, is present on chromosome VI of isolate RED_2019, whole genome shotgun sequence (position 809306–810340 negative strand). The AgB12_g028_t01 transcript codes for a protein of 405 amino acids with an isoelectric point of pH 3.98. The deduced Asu-EPIC-3 sequence has a signal peptide followed by five TRs of varying length (48 to 61 amino acids) with three tyrosine residues per repeat, except repeat1, which contains only two tyrosine residues. The amino acid sequence motifs differ from the other two epicuticlins in the conserved tyrosine-containing regions (KRNNE**YGD**EP and EIEAS**GYR**K) (Fig 6 and Dryad, Dataset, https://doi.org/10.5061/dryad.fttdz08vs). In the human parasite *A. lumbricoides*, a closely related species to *A. suum* [48], three different proteins (Uniprot accessions A0A0M3I7G7; A0A0M3IF25; A0A0M3I7G8), sharing high similarity with the epicuticlins of *A. suum* were identified (Fig 6; S2 Table). Compared to their homologous in *A. suum*, Alu-EPIC-1 lacks one repeat (probably repeat 5). Alu-EPIC-2 has only two repeats, and Alu-EPIC-3 has one repeat more than Asu-EPIC-3. Different epicuticlin sequences were also identified in *T. canis* (S2 Table). Some of these epicuticlins have up to 33 repeats (e.g., A0A0B2UZ77 and A0A0B2V0B7). Three epicuticlins have the characteristics typically found in the *Ascaris* species. Tca-EPIC-1 (A0A183TVS5) has a repeat length of up to 51 amino acids and up to three tyrosine residues per repeat. Tca-EPIC-2 (A0A183V326) has three repeats of 54 amino acids and only two tyrosine per repeat. Tca-EPIC-3 has up to 61 amino acids and three tyrosine per repeat (Fig 6).

### Clade IV Tylenchina: *Steinernema glaseri*

*S. glaseri* is a representative species of the clade Tylenchina where only two different epicuticlins could be identified (A0A1I8ANA2 and A0A1I7YLQ6). Both have a signal peptide followed by six repeats with 50 to 56 amino acids. A0A1I8ANA2 has two tyrosines per repeat, and A0A1I7YLQ6 has three tyrosines. (Fig 6 and S2 Table).

### Clade V Rhabditina: *Ancylostoma caninum*, *Pristionchus pacificus*, and *C. elegans*

Two different epicuticlin sequences were found in *A. caninum* (A0A368GWQ4 and A0A368GWR3) with nine and four repeats, respectively (Fig 6). A0A368GWQ4 has the particularity that one cysteine per repeat is present between the two tyrosine motifs. The last repeat has no cysteine. A0A368GWR3 has up to four tyrosines per repeat (Fig 6). The two epicuticlins of *P. pacificus* differ in the repeat number and length: A0A2A6C2N0 has six repeats of 52 to 69 amino acids, and H3DXI4 has thirteen repeats of 51 to 81 amino acids. The tyrosine numbers are different as well, ranging from two to five. No cysteine residues were found in the repeats of these epicuticlins.

In the best-studied nematode *C. elegans*, epicuticlins have not yet been published in peer-reviewed journals. A previous PCR screen in a *C. elegans* library recovered sequences resembling *Asu-epic-1* in genes F11E6.3 (Q9U3J8) and K08D12.6 (Q8MXU8) [30] (S2 Table). F11E6.3 gene is localized on chromosome IV (position 17470514–17471920, negative strand) and consists of two exons transcribed into a pre-messenger of 1199 nucleotides, containing a coding region of 1038 nucleotides. The encoded protein Q9U3J8 comprises 345 amino acids with an estimated molecular weight of 35.2 kD and an isoelectric point of 4.02. It contains six TRs with 53–58 amino acids (Fig 6). Each repeat includes the two tyrosine motifs within the N- and C-terminal conserved regions (KRQAQNS**YGD**EA and DA**GYR**S). Cysteine residues are absent in the repeats of this epicuticlin. The second epicuticlin, gene K08D12.6, is localized on chromosome IV (positions 1720790–1723072, forward strand). The deduced protein

Q8MXU8 has 668 amino acids, an estimated molecular weight of 62.8 kD, and an isoelectric point of 4.21. It contains ten TRs varying between 57 and 70 amino acids (Fig 6). The conserved motifs comprising two tyrosine residues (KRNA**YGD**EQVT and DS**GYR**S) differ from those of Q9U3J8. This epicuticlin contains one cysteine per repeat, except repeat eight. These two *C. elegans* epicuticlins are expressed in the larval stages (L1-L4) and in young adults with aggregate expression estimates of mean FPKM values (Fragments Per Kilobase of transcript per Million fragments mapped) from 119 to 590 in Q9U3J8 and from 96 to 486 for Q8MXU8. In dauer larvae and embryos, the expression is low.

STRING analysis of the epicuticlin Q8MXU8 indicated a co-expression and homology with the epicuticlin Q9U3J8 (Fig 7).

CELE_F46F2.3 and CELE_T04F8.8 are annotated as uncharacterized IDPs. The first, featuring 137 amino acids and 13 tyrosine residues, is expressed in the hypodermis. The second protein of 165 amino acids with 16 tyrosine residues is rich in proline. The location of its expression is unknown. Co-expressed cuticular collagens belong to the cuticular collagen

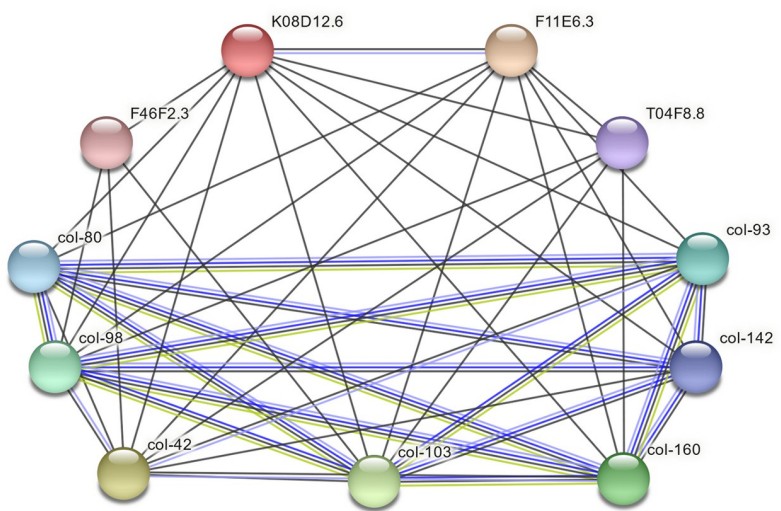

| Predicted Functional Partners of Cel-EPIC-x (K08D12.6): | | Coexpression | Homology | Score |
|---|---|---|---|---|
| F11E6.3 | Protein F11E6.3 (F11E6.3) mRNA, complete cds | + | + | 0.53 |
| col-42 | COLlagen; Protein Y69H2.14 (Y69H2.14) mRNA, complete cds | + | | 0.526 |
| col-103 | COLlagen; Protein COL-103 (col-103) mRNA, complete cds | + | | 0.523 |
| col-160 | COLlagen; Protein COL-160 (col-160) mRNA, complete cds | + | | 0.52 |
| col-98 | COLlagen; Protein COL-98 (col-98) mRNA, complete cds | + | | 0.515 |
| col-93 | COLlagen; Protein COL-93 (col-93) mRNA, complete cds | + | | 0.487 |
| col-80 | Putative cuticle collagen 80 | + | | 0.467 |
| col-142 | COLlagen; Protein COL-142 (col-142) mRNA, complete cds | + | | 0.463 |
| T04F8.8 | Protein T04F8.8, isoform b (T04F8.8) mRNA, complete cds | + | | 0.461 |
| F46F2.3 | Protein F46F2.3 (F46F2.3) mRNA, complete cds | + | | 0.454 |

**Fig 7. Scheme of STRING analysis of the *C. elegans* epicuticlin Q8MXU8 (K08D12.6) to show potential interactions of this epicuticlin with the second epicuticlin Q9U3J8 (F11E6.3), cuticular collagens, and other proteins rich in tyrosine.** The red-colored node represents the query protein, and the other nodes are the first shell of interactors. Co-expression and homology analysis are indicated in the Table. Each protein-protein interaction is annotated with 'scores', which are confidence indicators. All scores rank from 0 to 1, with 1 being the highest possible confidence. A score of 0.5 indicates that roughly every second interaction might be erroneous (i.e., a false positive) [39].

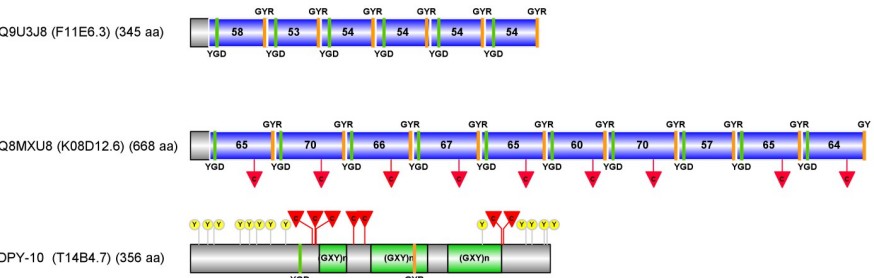

**Fig 8. Schematic representation of the two *C. elegans* epicuticlins Q9U3J8 and Q8MXU8 and the cuticular collagen DPY-10.** The Uniprot accession numbers, in parentheses, the Wormbase accession, and the total amino acid numbers are shown. The three protein sequences are drawn to scale. The TRs in the two epicuticlins are indicated as blue boxes with the amino acid numbers. The GYR and YGD motifs are shown as orange and green boxes, respectively. The positions of the cysteine residue in the epicuticlin Q8MXU8 and the eight cysteine residues, organized in three clusters (red triangles) in the collagen, are marked with red triangles. In the collagen, the three collagen (GXY)n regions (green boxes), the two tyrosine motifs, and thirteen tyrosine residues (yellow circles) are indicated.

clusters C (COL-142 and COL-93) or D [49]. As collagens are potential first interactors with epicuticlins, we looked for collagens potentially featuring YGD and GYR motifs. The collagens indicated in the String analysis had no such motifs. Among the 173 cuticular collagen proteins available in the *C. elegans* matrisome database [49], sixteen had one or two of these motifs (Dryad (https://doi.org/10.5061/dryad.fttdz08vs). The collagen DPY—10 of cluster B13 was the only one with YGD and GYR motifs (Fig 8).

A schematic representation of the two *C. elegans* epicuticlins and the DPY—10 collagen with their YGD and GYR motifs, their cysteine and tyrosine residues illustrates multiple potential partner sites which would be involved in the assembly of the epicuticlin layer with the cuticular collagens. The epicuticlin Q8MXU8 with the regularly spaced cysteine and the GYR and YGD motifs places it in privileged conditions to interact with the different tyrosine residues and the cysteine clusters present in the collagen, as well as with the YGD and GYR motifs of the Q9U3J8 epicuticlin.

## Discussion

### *A. suum*, a crucial starting point for the identification of epicuticlins

The isolation and purification of sufficient insoluble cuticular material is easy using *A. suum*, with a size of up to 20 cm [24]. Modified procedures were used to isolate insoluble fractions, for example, of the cuticles of *C. elegans* [15], *A. viteae* [50], and *B. pahangi* [51]. However, the insolubility and the small amounts of material obtained hampered their further biochemical characterization. Yet, [125]I labeling methods demonstrated that a significant part of the labeled tyrosine-containing proteins remained in the insoluble epicuticular fraction [52, 53]. Immunogold and lectin-gold techniques showed no reaction with the epicuticular outer surface [54]. However, specific antibodies against the isolated and purified insoluble epicuticular residue of *A. suum* allowed the isolation of a cDNA clone, X92101, encoding repetitive motifs rich in alanine, proline, and tyrosine, typical characteristics found in the epicuticle [29]. Due to the repetitive nature, the sequencing of the cDNA clone remained incomplete. This sequencing problem is also observed in several genomic and cDNA sequences deposited in databanks by different laboratories. For example, scaffold data (GCA 18702v3; PRJNA62057 or Asc-Suum_1.0; PRJNA80881) are incomplete in the assumed epicuticlin repeat regions (for example gene ID AgB12_g029 and GS_02605). Using ExoIII/Mung Bean nuclease allowed us to

overcome these problems [30] before its use for sequencing long stretches of repetitive DNA [55]. However, only the recent publication of the fully assembled chromosomes of *A. suum* (GCA 1343314v1) [31] made it possible to identify now the complete *Asu-epic-1* gene.

## Characteristics of the epicuticlin genes

The nucleotide repeats in the *Asu-epic-1* gene match with repeats of all previously identified sequences, although some differences appeared, such as nucleotide substitutions and variations in the repeat numbers. The identification of the unique nucleotide patterns allowed us to achieve an optimal manual alignment and comparison of the diverse epicuticlin repeat sequences. The variations in the repeat number in the cDNA and the genomic PCR products are mostly related to insertion/deletion (Indel) incidences since no alternative splicing is involved, and only a single copy of the gene was identified in the genome. Moreover, the fact that the epicuticlins are intrinsically disordered proteins and a higher Indel rate was described in disordered domain repeats [56] reinforces our suggestion concerning slippage of polymerases and Indel generations. Recombination events, such as unequal crossing over and gene conversion, may additionally lead to contractions and expansions of TR sequences by which repeat-containing haplotypes rapidly evolve. Similar nucleotide patterns are present in the nucleotide sequences of epicuticlins of other nematodes, but they have not been analyzed.

Asu-epic-1* and *Asu-epic-3* are relatively closely located on chromosome VI, and *Asu-epic-2* is on chromosome III. However, the two *C. elegans* epicuticlin genes (F11E6.3 and K08D12.6) are at the opposite ends of chromosome IV. Wang et al. [31] compared the chromosomes of both species, and a significant part of genes present on chromosome VI of *A. suum* have their homologs on chromosome IV of *C. elegans*. The exact evolutionary chromosomal relationship of the epicuticlins in both species remains to be shown. We could not detect epicuticlin sequences in Dorylaimia (Clade I) and Enoplia (Clade II). It is possibly due to incomplete sequencing data or the presence of another category of epicuticular proteins in these nematodes. Additional epicuticlins might be detected in several nematode species through an in-depth analysis of the available databanks and future whole-genome sequencing projects.

All identified epicuticlins are characterized by varying numbers of TRs. The number of repeats in *Asu-epic-1* and the two *C. elegans* epicuticlins was ascertained through the available DNA gene sequences. However, the repeat numbers of epicuticlin-like sequences of many other species should be re-examined. The very high numbers of TRs (>30) found in the sequences of *T. canis* (KHN74524.1 and KHN74525.1) were surprising. However, we found identical sequences to these proteins (VDM24067.1 and VDM24068.1), which have only nine and three TRs, respectively. KHN74524.1 (32 TRs) and VDM24067.1 (9 TRs) are hypothetical proteins deposited in databanks of two different bioprojects (PRJEB533 and PRJNA248777) carried out by two different research groups. Despite the difference in the repeat numbers, their gene sequences (with the flanking, upstream, and downstream untranslated regions) are practically identical. The discrepancy in the repeat numbers could be due to the sequencing and assembly difficulties associated with the TRs. Again, more accurate sequencing/assembly data are needed to determine the actual repeat number.

## Epicuticlins are expressed in different larval stages of the model nematode *C. elegans*

The cuticle of nematodes is newly synthesized during the four molting processes from L1—L4 larval stages to the adult stage [57]. The complexity of the processes involved has been recently reviewed in *C. elegans* [58]. Correct temporal expression of the genes and the synthesis of cuticular proteins are crucial. While oscillating gene expression has been shown for the

cuticular collagen genes of *C. elegans* [59], it is not known whether the epicuticlin expression follows a similar temporal oscillation pattern. However, it is evident that the expression levels of the *C. elegans* epicuticlin genes F11E6.3 and K08D12.6 increase significantly through all larval stages to young adults. At the same time, it is absent or low in the early embryo and dauer stages. F11E6.3, K08D12.6 and K08D12.7 were recently annotated in Wormbase as *epic-1*, *epic-2* and *epic-3*. Since the epicuticlins are involved in building the outermost electron-dense layer of the cuticle, we propose that their synthesis and secretion start somewhat before the collagens are synthesized. The recently described role of TMEM131 proteins for the cargo collagen secretion in *C. elegans* [60] will undoubtedly help to understand the secretion machinery and let us suggest that a similar cargo procedure may secrete the epicuticlins.

## Epicuticlins, as intrinsically disordered proteins with tandem repeats and molecular recognition features, are prime candidates as nucleation sites for the formation of the cuticle

The presumed Asu-EPIC-1 protein corresponds in its biochemical properties to the insoluble residue originally described by Fujimoto and Kanaya [24]. Several attempts to produce recombinant epicuticlin proteins using a wide variety of different expression systems were unsuccessful. So far, the only expression was possible via the production of a fusion protein using the 26-kDa GST tag from *Schistosoma japonicum* [61] subcloned into the plasmid vector pGEX [29]. This failure might be associated with the intrinsic characteristics of the epicuticlins since other recombinant nematode proteins were simultaneously successfully produced [62].

All the analyzed epicuticlins are predicted to be IDPs, apart from their signal peptide. Intrinsically disordered proteins (IDP) have gained increasing attention during the last decade [44, 63, 64]. Their identification as weak polyampholytes, showing structures of globule and tadpole-like proteins [42], could point to their crucial role in assembling the cuticles. The combination of intrinsically disordered regions (IDR) and tandem repeats is frequent, and Delucchi et al. [65] defined four categories of TRs, according to the repeat unit length. Repeats with 15 or more residues were called domain repeats, representing 30% of all predicted TR. The association of TR with intrinsic disorder allowed the distinction of four types of overlaps. The epicuticlins are IDPs with tandem repeats of the category of domain repeats (> 15 residues) and belong to the fraction of proteins where IDR and TR cover the whole protein length. The analyzed epicuticlin sequences are predicted to have molecular recognition features (MoRFs), short protein-binding regions that undergo disorder-to-order transitions upon binding appropriate protein partners [66, 67]. We propose that these properties of epicuticlins enable them to be disorder-enriched hubs for protein-protein linking [65, 68] in the cuticle assembly. They could play essential roles in interacting with other epicuticlins, different cuticular collagens, cuticlins, and glycoproteins. The STRING analysis indicates that the cuticular collagens are privileged partners for such interactions, probably due to tyrosine and cysteine clusters. Based on the rotary shadowing of collagen molecules, the length of cuticular collagens of *A. suum* was calculated to be 45 nm [69]. The theoretical size of Asu-EPIC-1, assuming a linear protein with 470 amino acids, would be around 112 nm. Interaction of two canonical collagen units with one epicuticlin protein is theoretically possible via the proline, tyrosine, or cysteine residues.

## Roles of conserved motifs of epicuticlins and their homologs in arthropods

The epicuticlins contain between six and ten repeats. Whether this represents a fixed pattern is not clear. If the number of repeats should be vital in a structural crosslinking pattern, a specific number of repeats could be expected to perform a given function. Remarkably, one tyrosine is

present in the two conserved motifs of all epicuticlins of all nematode species. One or more additional tyrosine residues is/are present in the repeats outside the two conserved motifs in several epicuticlins. Each of these additional tyrosine residues is present within a potential not-annotated motif. In general, tyrosine residues are well known to be involved in crosslinking processes [18, 70]. In *A. suum*, dityrosine was identified in the hydrolysate of cuticlin [71] and the cuticle of larvae of *H. contortus* [72]. Marti [25] isolated di-, tri-, and iso- trityrosine from the hydrolyzed cuticlin of *A. suum* and showed a characteristic blue fluorescence of cuticlin under UV excitation at 254 nm, typical for dityrosine. Specific antibodies raised against dityrosine labeled the electron-dense layers of the epicuticlin, confirming the presence of dityrosine in the insoluble epicuticular structures. The tyrosine residues identified in the conserved epicuticlin motifs GYR and YGD are the prime sites in forming dityrosine and linking epicuticlins and other cuticular proteins. The role of the 29-kDa glycoprotein (gp29) identified in the cuticles of filarial nematodes [13] in catalyzing the formation of the crosslinking residues should be further analyzed. Several nematode species of the clade V have two different epicuticlin types, distinguished via the presence or absence of cysteine residues in the repeats. The cysteines are localized between the conserved motifs GYR and YGD. We propose that they could also be involved in the crosslinking with cysteines in the cuticular collagens.

The conserved Pfam annotated (PF02756) GYR motif is present in cuticular protein families of arthropods [46, 73]. Twelve cuticular protein (CP) families were defined by Willis [74], and they were suggested to be involved in the arthropod cuticle assembly. Willis mentioned: "All of these protein families appear to be restricted to arthropods. . ." and "Furthermore, given the similar cuticle construction throughout the Ecdysozoa (Schmidt-Rhaesa et al., 1998), it is intriguing that the arthropods seem to have adopted so many unique configurations for their CPs". Detecting the GYR and YGD motifs in the nematode epicuticlins made it possible to compare them with the motifs in the Tweedle and CPR families of arthropods [46]. For the first time, the presence of similarities between cuticular proteins of arthropods and nematodes is confirmed. Whereas in arthropods, motif GYR frequently is located within the N-terminal region and the motif YLP within the C-terminal region, the selected *D. grimshawi* cuticular protein B4IZ60 has in each repeat first the motif YLP, followed by the motif GYR. The role of the additional tyrosine residue motifs in arthropod and nematode proteins remains to be clarified. New ways are now opened to study the formation of the cuticular structures using proteomics and matrisomic approaches [49]. A combination of diverse genetic and molecular techniques well established for *Drosophila* and *Caenorhabditis* model organisms will be helpful.

## Outlook

Innovative ways to produce epicuticlins are needed. Recombinant epicuticlins would be especially useful for future applications in the biomaterial sciences, as shown for recombinant resilin [75] and elastin [76]. The potential of the highly organized characteristics of epicuticlins is evident, for example, as biosynthetic polymers in medical applications, such as in skin healing procedures. Of general scientific interest is the clarification of the synthetic machinery leading to the formation of the complex extracellular matrix of the cuticle of nematodes and arthropods. In general, even if the role of repetitive sequences is more and more recognized, this study underlines their importance.

## Supporting information

**S1 Fig. Clustal alignment of the 5' regions of *Asu-epic-1* genomic and cDNA sequences.** (DOCX)

**S2 Fig. Scheme of the identification of the complete *Asu-epic-1* gene of *A. suum*.**
(DOCX)

**S3 Fig. Comparative alignment of the amino acid sequence of Asu-EPIC-1 with sequences encoded by four cDNA and one TSA clone.**
(DOCX)

**S1 Table. BLASTN results with query AJ408887, an *A. suum* partial mRNA.**
(DOCX)

**S2 Table. Epicuticlin sequences detected in different nematode species.**
(DOCX)

# Acknowledgments

We would particularly thank all the former Ph.D., MSc students, and technicians who contributed to elaborating some of the initial data.

# Author Contributions

**Conceptualization:** Bruno Betschart, Marco Bisoffi, Ferial Alaeddine.

**Data curation:** Bruno Betschart.

**Formal analysis:** Bruno Betschart, Ferial Alaeddine.

**Funding acquisition:** Bruno Betschart.

**Investigation:** Bruno Betschart, Marco Bisoffi, Ferial Alaeddine.

**Methodology:** Bruno Betschart.

**Project administration:** Bruno Betschart.

**Supervision:** Bruno Betschart.

**Validation:** Bruno Betschart.

**Writing – original draft:** Bruno Betschart, Marco Bisoffi, Ferial Alaeddine.

**Writing – review & editing:** Marco Bisoffi, Ferial Alaeddine.

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
