## [Decision Letter · Decision Letter 0]

29 Jun 2022

PONE-D-22-12806Identification and characterization of epicuticular proteins of nematodes sharing motifs with cuticular proteins of arthropodsPLOS ONE

Dear Dr. BETSCHART,

Thank you for submitting your manuscript to PLOS ONE. After careful consideration, we feel that it has merit but does not fully meet PLOS ONE’s publication criteria as it currently stands. Therefore, we invite you to submit a revised version of the manuscript that addresses the points raised during the review process.

We look forward to receiving your revised manuscript.

Kind regards,

Denis Dupuy, Ph.D.

Academic Editor

PLOS ONE

Journal Requirements:

2. Please amend the manuscript submission data (via Edit Submission) to include author “Marco Bisoffi, Ferial Alaeddine”

Reviewers' comments:

Reviewer's Responses to Questions

**Comments to the Author**

1. Is the manuscript technically sound, and do the data support the conclusions?

Reviewer #1: Yes

Reviewer #2: Yes

2. Has the statistical analysis been performed appropriately and rigorously? 

Reviewer #1: Yes

Reviewer #2: N/A

3. Have the authors made all data underlying the findings in their manuscript fully available?

Reviewer #1: No

Reviewer #2: Yes

4. Is the manuscript presented in an intelligible fashion and written in standard English?

Reviewer #1: No

Reviewer #2: Yes

5. Review Comments to the Author

Reviewer #1: Overview: The authors annotate a small gene family implicated as a structural component of the nematode epicuticle. They determine the proteins to have a highly repetitive, intrinsically disordered structure similar to other cuticular-protein families, including similar composition biases and repeated tyrosine-associated motifs. While no quantitative gene expression data are as yet provided, the authors do show that a C. elegans homolog is predicted to interact with other cuticular components such as collagens based on co-expression and other evidence. The study provides useful information that culminates extensive previous work in characterizing nematode cuticular proteins. My main complaints relate to the excessive length throughout, including some convoluted language and superfluous detail (in the figures as well). Paragraphs could be better structured to order the logical flow of ideas. Embedding the figure legends directly in the text was an obstacle to understanding and reviewing their work, please remove them to a terminal section in a revision.

I consider most of the suggested changes minor. However, the overall need to shorten and better focus the manuscript will require substantial re-writing. Therefore I recommend “major revisions”.

Asuum epicuticlin review:

Major comments

1. Both the abstract and introduction are too long and contain extraneous detail for this audience.

2. Embedding of figure legends in the text - not sure if this is requested by the journal but it is confusing in a reviewer pdf.

3. Paragraphs could be more structured with topic/transition sentences. They tend to run on and are intermixed with figure legends, obscuring the logical sequence.

Minor comments

1. Line 31 “is composed of an insoluble protein called epicuticlin”. This seems to assert that there is only one epicuticlin, which seems contrary to the current work and generally not discernable from proteomic analysis.

2. Line 41: “Non-canonical trajectories” is excessively vague. Suggest changing “Non-canonical trajectories seem to be followed by the polymerases” to “variation in repeat number was observed in both genomic and mRNA sequences, potentially due to replication slippage”? This is a well-known molecular mechanism that generates variation in simple repeats, such as glutamine tracts or noncoding microsatellites, as you more clearly point out in the discussion. It may also be worth pointing out (in the Discussion) that unequal crossing over and gene conversion are common mechanisms by which repeat-containing haplotypes rapidly evolve.

3. Lines 113-123 are unnecessary: I agree it is important to explain the state of knowledge linking cDNA or genomic clones to proteins localized in the nematode cuticle (challenging work), but suffice it to say in the intro that no full length gene structure could be inferred prior to genomic sequencing. This is a familiar problem and does not require this detailed review of splicing ambiguity.

4. Line 48: “Asu-EPICUT1 is an intrinsically disordered protein (IDP) having six molecular recognition features (MoRFs).” Insert “predicted” after “six”. These are computationally predicted features with uncertainty associated. The MoRF analysis is described in the results and instead should be described in the Methods.

5. Line 96: Use of “contrary” is grammatically incorrect. Suggest “In contrast, specific antibodies raised against the purified A. suum cuticlin reacted within the electron-dense, insoluble, outermost layer and filarial epicuticular structures” or similar.

6. Line 98: delete “detailed” as redundant

7. Analyses listed in the Methods are not referenced in the rest of the paper, e.g. use of Compute and ProtParam

8. Line 175: Changing “using unidirectionally deleted fragments produced” to “after blunting amplicons with” might make purpose of exonuclease treatment clearer? Move sentence to methods regardless.

9. Line 179: try to be explicit when stating %identity whether the comparison is nucleotide or protein. Not always obvious by context.

10. Line 192: the countries listed are not relevant to interpretation

11. Line 192: Change “The rest of the not yet annotated genomic hits were sequences” to “The remaining hits were unannotated sequences”

12. Lines 187-197 seem unnecessary. Readers do not want to read about a blast result that can be [and is!] summarized in a table. It seems sufficient to state that JI176387.1 was identified as a representative mRNA from which to build an initial gene model, although this is a TSA sequence and not WGS correct?

13. Line 202: Change “have been” to “were” and move whole sentence to methods.

14. Line 204: It was not clear at this point whether the authors were referring to an existing hypothetical gene prediction or an entirely new annotation of their own. It becomes clearer later, but confusing here. Suggest “The proposed gene” or similar.

15. Line 206 Verbose: “Comparison of the gene and the cDNA JI176387.1 confirms that the pre-mRNA is submitted to a cis-splicing process” could simply be “Alignment of cDNA JI176387.1 to the gene model confirmed the predicted exon structure”

16. Lines 205 and 213 repeat the chromosomal location.

17. Figure 2a seems supplemental to me at best. There are simpler ways to summarize a cDNA alignment. It is unnecessary to produce a figure with extensive nucleotide sequence written out in the main body of the paper.

18. More broadly, Figs 1-2B could be combined and streamlined to show the first ideogram of Fig.3 (which as color coded implies the next two ideograms), together with aligned evidence more succinctly displayed (individual clones as line tracks).

19. I think using the “motif1” and “motif2” nomenclature of Cornman is unhelpful here. That study generated a set of motifs de novo, hence the nomenclature, and then subsequently determined that they overlapped with GYR and YLP Pfam motifs.

20. It remains true that GYR and YLP hmms in Pfam are strongly Drosophila/Diptera focused, it’s not clear how much of that is search bias, or overtraining the very short HMMs to Dipteran sequence.

21. Line 454: “a very close species of A. suum” is grammatically incorrect. “a species closely related to A. suum”. Support for that statement should be referenced.

22. Line 489: “PCR in a C. elegans library allowed detecting sequences resembling Asu-epicut1” is grammatically incorrect. Suggest combining such as “A previous PCR screen of a C. elegans library recovered sequences resembling Asu-epicut1”.

23. The first two paragraphs of the discussion repeat unnecessary detail of already published work. They also repeat text already present in the manuscript elsewhere.

24. I do not see any value in reviewing nucleotide repeats in coding sequence that is itself repetitive. The latter dictates the former. It is only helpful in identifying evolutionary processes underlying repeat evolution.

25. Line 639: “repeat numbers is due” to “repeat numbers may be due”.

26. Line 643 Grammar: “and due the uniqueness of the gene in the genome suggests that”. Change to “and only a single copy of the gene was identified in the genome,”.

27. Line 647 replace “showed to be” with “are”.

28. Line 651 “Cuticlins and epicuticlins are different types of proteins” seems a tautology. Change to something like “Epicuticlins have characteristics distinct from cuticlins”?

29. Line 736: Reference [20] is Lewis not Marti

30. Line 780-781: Presumably the tyrosine crosslink motifs would co-vary with the actual enzymes that do the crosslinking.

31. Figure 4 should be substantially reduced, in part by removing the EST sequences. ESTs are often fragments or of poor quality, and their short length here limits any contribution to the overall interpretation. Part A is more useful in demonstrating TR variability, Part B can be dispensed with.

Reviewer #2: This is an interesting paper that identifies a family of epicuticular proteins in several nematode species and helps to clarify how the nematode cuticle is put together. Some comments and questions:

Line 26: In the abstract, the sentence “Since nematodes are found in aquatic, marine, and soil environments and as parasites in plants or animals, the cuticle is a highly protective structure” suggests that if they were only found in e.g., the soil the cuticle would not be a highly protective structure? Reword.

41: I do not understand what “Non-canonical trajectories” means. Explain.

50: Write: “motifs” not “motives”

56: Define “TR’ here.

81: For context, can the authors say a little about the non-collagen cuticle glycoproteins; what do we know about them?

92: Say more about CUT-1 here and the relationship between CUT-1 and CUT-2.

141: What does “(aagaggaa)” signify here? Is this the conserved domain?

156: Define IDP.

In table S1, why note the “Country”? Delete?

Explain what “Bits” refers to, in the S1 table. Does “% positives” mean similarity/identity? Define “resp score”.

Reword the S1 table text since “mRNA coding for the epicuticlin gene” does not make sense.

318: I cannot see the motifs mentioned the sequence presented in figure 3?

326: There is no figure 5E.

365: What exactly are the Drosophila matches, mentioned here?

404: Incomplete sentence about raw data?

443: Better to underline the five. Italics are hard to discern.

703: Write “were” instead of “showed to be”.

764: insert “in” between “present” and “the”.

The point of figure S1 is unclear. The genomic and cDNA sequences do not align; what are the authors showing us here?

Likewise, the relevance of figure S2 escapes me. For instance, in repeat 1 at base 12 there is a “t”. Is this the only variation seen at this position? In how many cases? Explain.

In figure 8, its not immediately obvious which node is K08D12.6. Point out in the legend that this is the red node. Only some of the interactors are listed below the node scheme; what about the rest?

6. PLOS authors have the option to publish the peer review history of their article (what does this mean?). If published, this will include your full peer review and any attached files.

Reviewer #1: No

Reviewer #2: No

---

## [Author Response · Author response to Decision Letter 0]

11 Aug 2022

We want to thank the reviewers for their helpful critical evaluation of our manuscript. The comments allowed to improve the text considerably. 

In general, we streamlined a lot:

• The abstract was reduced from 519 to 332 words

• The Introduction was reduced from 866 to 755 words

• The M&M was increased from 242 to 327 words, mainly due to a transfer of information from results section to M&M.

• The result section (text without figure legends) was reduced from 3087 to 2320 words

• The discussion was reduced from 2704 to 1941 words

• The number of figures was reduced from 9 to 8, and the figures were redesigned and simplified. The legends were simplified as well.

Despite the streamlining, the additional information asked for by the reviewers was included.

Detailed comments:

Academic editor: Author’s comments

Is provided

A marked-up copy of your manuscript that highlights changes made to the original version. You should upload this as a separate file labeled 'Revised Manuscript with Track Changes'. Is provided

Is provided

Has been verified and corresponds to PLOS ONE requirements

2. Please amend the manuscript submission data (via Edit Submission) to include author “Marco Bisoffi, Ferial Alaeddine” 

Both author names are included

We have removed and reformulated the corresponding text

4. We note that you have stated that you will provide repository information for your data at acceptance. Should your manuscript be accepted for publication, we will hold it until you provide the relevant accession numbers or DOIs necessary to access your data. If you wish to make changes to your Data Availability statement, please describe these changes in your cover letter and we will update your Data Availability statement to reflect the information you provide. This has been done and is included in the cover letter as information as well

Comments to the authors: 

1. Is the manuscript technically sound, and do the data support the conclusions?

Reviewer #1: Yes 

Reviewer #2: Yes 

2. Has the statistical analysis been performed appropriately and rigorously? 

Reviewer #1: Yes 

Reviewer #2: N/A 

3. Have the authors made all data underlying the findings in their manuscript fully available?

Reviewer #1: No The data availability information (NCBI accession numbers and Dryad data accession) is given in the text and in the cover letter

Reviewer #2: Yes 

4. Is the manuscript presented in an intelligible fashion and written in standard English?

Reviewer #1: No The manuscript has been reformulated, streamlined, restructured, and checked with Grammarly software for typing or grammatical errors

Reviewer #2: Yes 

Reviewer 1: 

Overview: The authors annotate a small gene family implicated as a structural component of the nematode epicuticle. They determine the proteins to have a highly repetitive, intrinsically disordered structure similar to other cuticular-protein families, including similar composition biases and repeated tyrosine-associated motifs. While no quantitative gene expression data are as yet provided, the authors do show that a C. elegans homolog is predicted to interact with other cuticular components such as collagens based on co-expression and other evidence. The study provides useful information that culminates extensive previous work in characterizing nematode cuticular proteins. My main complaints relate to the excessive length throughout, including some convoluted language and superfluous detail (in the figures as well). Paragraphs could be better structured to order the logical flow of ideas. Embedding the figure legends directly in the text was an obstacle to understanding and reviewing their work, please remove them to a terminal section in a revision. Gene expression data are given for the epicuticlin genes of C.elegans and the EST and TSA data of A.suum confirm the expression of the gene even if no quantitative data are available

I consider most of the suggested changes minor. However, the overall need to shorten and better focus the manuscript will require substantial re-writing. Therefore I recommend “major revisions”. 

The length of the manuscript has been significantly reduced and the text was focused (details are given above) 

Embedding the figure legends is according to the guidelines of PLOS ONE. We provide therefore three different versions of the manuscript: 

According to PLOS One guidelines with or without track changes and as asked for by reviewer 1 a separate file (Others) with figure legends as a separate section at the end of the manuscript

Major comments 

1. Both the abstract and introduction are too long and contain extraneous detail for this audience. 

Streamlined and shortened 

2. Embedding of figure legends in the text - not sure if this is requested by the journal but it is confusing in a reviewer pdf.

See above and PLOS ONE: Place figure captions in the manuscript text in read order, immediately following the paragraph where the figure is first cited. Do not include captions as part of the figure files or submit them in a separate document. 

3. Paragraphs could be more structured with topic/transition sentences. They tend to run on and are intermixed with figure legends, obscuring the logical sequence.

We better structured the text 

Minor comments 

1. Line 31 “is composed of an insoluble protein called epicuticlin”. This seems to assert that there is only one epicuticlin, which seems contrary to the current work and generally not discernable from proteomic analysis.

New text: is composed of insoluble proteins called epicuticlins

2. Line 41: “Non-canonical trajectories” is excessively vague. Suggest changing “Non-canonical trajectories seem to be followed by the polymerases” to “variation in repeat number was observed in both genomic and mRNA sequences, potentially due to replication slippage”? This is a well-known molecular mechanism that generates variation in simple repeats, such as glutamine tracts or noncoding microsatellites, as you more clearly point out in the discussion. It may also be worth pointing out (in the Discussion) that unequal crossing over and gene conversion are common mechanisms by which repeat-containing haplotypes rapidly evolve. 

The phrasing was changed accordingly. Yet it is not only the number of repeats which changes but also the type of repeats. In discussion added:

Also, recombination events, such as unequal crossing over and gene conversion may additionally lead to contractions and expansions of TR sequences by which repeat-containing haplotypes rapidly evolve.

3. Lines 113-123 are unnecessary: I agree it is important to explain the state of knowledge linking cDNA or genomic clones to proteins localized in the nematode cuticle (challenging work), but suffice it to say in the intro that no full length gene structure could be inferred prior to genomic sequencing. This is a familiar problem and does not require this detailed review of splicing ambiguity. 

New text: 

No full length gene structure could be inferred prior to genomic sequencing

4. Line 48: “Asu-EPICUT1 is an intrinsically disordered protein (IDP) having six molecular recognition features (MoRFs).” Insert “predicted” after “six”. These are computationally predicted features with uncertainty associated. The MoRF analysis is described in the results and instead should be described in the Methods.

New text: 

six predicted….

5. Line 96: Use of “contrary” is grammatically incorrect. Suggest “In contrast, specific antibodies raised against the purified A. suum cuticlin reacted within the electron-dense, insoluble, outermost layer and filarial epicuticular structures” or similar.

New text: 

In contrast, specific antibodies raised against the purified A. suum cuticlin reacted with the electron-dense insoluble outermost layer of A.suum and filarial epicuticular structures 

6. Line 98: delete “detailed” as redundant

Was deleted

7. Analyses listed in the Methods are not referenced in the rest of the paper, e.g. use of Compute and ProtParam

Were inserted or deleted. 

8. Line 175: Changing “using unidirectionally deleted fragments produced” to “after blunting amplicons with” might make purpose of exonuclease treatment clearer? Move sentence to methods regardless. 

We changed the formulation by shortening the phrase and keeping the reference only. There was no need to transfer it in M&M section. It was a method used in a former analysis. 

9. Line 179: try to be explicit when stating %identity whether the comparison is nucleotide or protein. Not always obvious by context.

Was done

10. Line 192: the countries listed are not relevant to interpretation

Countries are deleted

11. Line 192: Change “The rest of the not yet annotated genomic hits were sequences” to “The remaining hits were unannotated sequences” 

The remaining hits were unannotated sequences of different whole- genome shotgun databases

12. Lines 187-197 seem unnecessary. Readers do not want to read about a blast result that can be [and is!] summarized in a table. It seems sufficient to state that JI176387.1 was identified as a representative mRNA from which to build an initial gene model, although this is a TSA sequence and not WGS correct? 

Lines deleted and rephrased to: The sequence JI176387rc was identified as a representative mRNA from which to build an initial gene model (S2 Figure). Yes it is a TSA sequence.

13. Line 202: Change “have been” to “were” and move whole sentence to methods.

The phrase became obsolete and has been deleted completely 

14. Line 204: It was not clear at this point whether the authors were referring to an existing hypothetical gene prediction or an entirely new annotation of their own. It becomes clearer later, but confusing here. Suggest “The proposed gene” or similar.

New text: 

The proposed gene

15. Line 206 Verbose: “Comparison of the gene and the cDNA JI176387.1 confirms that the pre-mRNA is submitted to a cis-splicing process” could simply be “Alignment of cDNA JI176387.1 to the gene model confirmed the predicted exon structure” 

Was changed

16. Lines 205 and 213 repeat the chromosomal location. 

Only once mentioned, since the new figure 1 legend does no longer mention this info

17. Figure 2a seems supplemental to me at best. There are simpler ways to summarize a cDNA alignment. It is unnecessary to produce a figure with extensive nucleotide sequence written out in the main body of the paper.

Figure 2 a has been transferred to supporting information. We are convinced that the data presented in the figure are important for an easier follow up of the work done

18. More broadly, Figs 1-2B could be combined and streamlined to show the first ideogram of Fig.3 (which as color coded implies the next two ideograms), together with aligned evidence more succinctly displayed (individual clones as line tracks). 

Has been simplified and streamlined accordingly

19. I think using the “motif1” and “motif2” nomenclature of Cornman is unhelpful here. That study generated a set of motifs de novo, hence the nomenclature, and then subsequently determined that they overlapped with GYR and YLP Pfam motifs.

The proposition is correct and we deleted motif1 and motif2 throughout the text, which led to a simplification of the text

20. It remains true that GYR and YLP hmms in Pfam are strongly Drosophila/Diptera focused, it’s not clear how much of that is search bias, or overtraining the very short HMMs to Dipteran sequence. 

This possibility exists, but we do not see the necessity to elaborate this point in the manuscript

21. Line 454: “a very close species of A. suum” is grammatically incorrect. “a species closely related to A. suum”. Support for that statement should be referenced.

Was changed and the reference was included

22. Line 489: “PCR in a C. elegans library allowed detecting sequences resembling Asu-epicut1” is grammatically incorrect. Suggest combining such as “A previous PCR screen of a C. elegans library recovered sequences resembling Asu-epicut1”.

Was changed

23. The first two paragraphs of the discussion repeat unnecessary detail of already published work. They also repeat text already present in the manuscript elsewhere.

Was revised and shortened

24. I do not see any value in reviewing nucleotide repeats in coding sequence that is itself repetitive. The latter dictates the former. It is only helpful in identifying evolutionary processes underlying repeat evolution.

We agree that a coding sequence is a consequence of the nucleotide sequence. The point here is to show that the nucleotide variations do not result in a modification of the amino acid sequences. This indicates an important role of the amino acid sequence in the function of the epicuticlins and limits evolutionary variations. We tried to formulate this point more precisely. See also point 31

25. Line 639: “repeat numbers is due” to “repeat numbers may be due”.

Was changed

26. Line 643 Grammar: “and due the uniqueness of the gene in the genome suggests that”. Change to “and only a single copy of the gene was identified in the genome,”.

Was changed

27. Line 647 replace “showed to be” with “are”.

Was changed

28. Line 651 “Cuticlins and epicuticlins are different types of proteins” seems a tautology. Change to something like “Epicuticlins have characteristics distinct from cuticlins”?

Was changed

29. Line 736: Reference [20] is Lewis not Marti

Was corrected and all references were reverified

30. Line 780-781: Presumably the tyrosine crosslink motifs would co-vary with the actual enzymes that do the crosslinking.

We have no information about this crucial point, but the tyrosine motifs are remarkably conserved in the epicuticlins

31. Figure 4 should be substantially reduced, in part by removing the EST sequences. ESTs are often fragments or of poor quality, and their short length here limits any contribution to the overall interpretation. Part A is more useful in demonstrating TR variability, Part B can be dispensed with.

EST sequences have been eliminated and 4b has been transferred into Supporting information. It was interesting to see that despite the nucleotide differences in the repeats the protein repeats are not changed drastically. 

Reviewer 2: 

This is an interesting paper that identifies a family of epicuticular proteins in several nematode species and helps to clarify how the nematode cuticle is put together. Some comments and questions: 

Line 26: In the abstract, the sentence “Since nematodes are found in aquatic, marine, and soil environments and as parasites in plants or animals, the cuticle is a highly protective structure” suggests that if they were only found in e.g., the soil the cuticle would not be a highly protective structure? Reword.

The phrase was deleted as it is unnecessary in this context and to shorten the abstract

41: I do not understand what “Non-canonical trajectories” means. Explain.

Has been changed to make it clear according to reviewers 1 suggestion

50: Write: “motifs” not “motives”

Was corrected to motifs

56: Define “TR’ here.

Is already defined in line 46(original version)

81: For context, can the authors say a little about the non-collagen cuticle glycoproteins; what do we know about them?

The corresponding text has been adapted and additional information is given by focusing mainly on the distinction between intrinsic and surface-associated glycoproteins. The proposed role of gp29 is outlined and is taken up in the discussion again

92: Say more about CUT-1 here and the relationship between CUT-1 and CUT-2.

In the introduction, the properties of cut-1 to cut-6 are summarized and the differences between cut-1 and cut-2 are mentioned. This allowed eliminating the corresponding paragraph in the discussion section

141: What does “(aagaggaa)” signify here? Is this the conserved domain?

Was clarified and reformulated

156: Define IDP.

Already made in abstract 

In table S1, why note the “Country”? Delete?

Was deleted

Explain what “Bits” refers to, in the S1 table. Does “% positives” mean similarity/identity? Define “resp score”. 

S1 Table was harmonized and the Bits, scores and % positives resp identity clarified

Reword the S1 table text since “mRNA coding for the epicuticlin gene” does not make sense.

Was reformulated

318: I cannot see the motifs mentioned the sequence presented in figure 3? 

The hydrophilic amino acid stretches RKKRNN are now indicated as red underlined stretches in Fig 3

326: There is no figure 5E. Thank you. 

Was lost during the processing. Is added and the complete figure 5 was streamlined

365: What exactly are the Drosophila matches, mentioned here?

Information is given: The horizontal lines indicate the sequences of the motifs corresponding most frequently to all Drosophila matches from the Interpro web site (http://www.ebi.ac.uk/interpro/) 

404: Incomplete sentence about raw data? 

Dryad data accession information is given

443: Better to underline the five. Italics are hard to discern. 

Underlined

703: Write “were” instead of “showed to be”.

Changed

764: insert “in” between “present” and “the”.

Added

The point of figure S1 is unclear. The genomic and cDNA sequences do not align; what are the authors showing us here? 

The text was reformulated to outline the relevance of this figure for the identification of Exon 1 and Exon 2

Likewise, the relevance of figure S2 escapes me. For instance, in repeat 1 at base 12 there is a “t”. Is this the only variation seen at this position? In how many cases? Explain. 

The figure became obsolete since we have redesigned Figure 2 completely

In figure 8, its not immediately obvious which node is K08D12.6. Point out in the legend that this is the red node. Only some of the interactors are listed below the node scheme; what about the rest? 

Figure 8 is redesigned and the interactors are limited to the first shell interactors summarized in the table.

---

## [Decision Letter · Decision Letter 1]

24 Aug 2022

PONE-D-22-12806R1Identification and characterization of epicuticular proteins of nematodes sharing motifs with cuticular proteins of arthropodsPLOS ONE

Dear Dr. BETSCHART,

Thank you for submitting your manuscript to PLOS ONE. After careful consideration, we feel that it has merit but does not fully meet PLOS ONE’s publication criteria as it currently stands. Therefore, we invite you to submit a revised version of the manuscript that addresses the points raised during the review process.

We look forward to receiving your revised manuscript.

Kind regards,

Denis Dupuy, Ph.D.

Academic Editor

PLOS ONE

Journal Requirements:

Reviewers' comments:

Reviewer's Responses to Questions

**Comments to the Author**

1. If the authors have adequately addressed your comments raised in a previous round of review and you feel that this manuscript is now acceptable for publication, you may indicate that here to bypass the “Comments to the Author” section, enter your conflict of interest statement in the “Confidential to Editor” section, and submit your "Accept" recommendation.

Reviewer #1: (No Response)

2. Is the manuscript technically sound, and do the data support the conclusions?

Reviewer #1: Yes

3. Has the statistical analysis been performed appropriately and rigorously? 

Reviewer #1: Yes

4. Have the authors made all data underlying the findings in their manuscript fully available?

Reviewer #1: Yes

5. Is the manuscript presented in an intelligible fashion and written in standard English?

Reviewer #1: Yes

6. Review Comments to the Author

Reviewer #1: The authors have adequately addressed my previous comments. However, their discussion of two epicuticlins in C. elegans does not seem to match what is currently annotated in WormBase. There are three C. elegans epicuticlins annotated with different nomenclature than that used by these authors: EPIC-1, EPIC-2, and EPIC-3. The annotations appear to be relatively recent (2022), so likely this is a new development since the authors began their study. I recommend updating the references to C. elegans epicuticlin genes and adopting the curated nomenclature in WormBase to minimize confusion.

7. PLOS authors have the option to publish the peer review history of their article (what does this mean?). If published, this will include your full peer review and any attached files.

Reviewer #1: No

---

## [Author Response · Author response to Decision Letter 1]

2 Sep 2022

Academic editor:

A rebuttal letter that responds to each point raised by the academic editor and reviewer is provided

A marked-up copy of the manuscript that highlights changes made to the original version is uploaded as a separate file labeled 'Revised Manuscript with Track Changes'. 

An unmarked version of the revised paper without tracked changes is provided.

Reviewer #1

To comply with the Wormbase nomenclature, recommended by Prof T. Schedl, whom we contacted, we decided to change our names from “epicut” to “epic” for all epicuticlin genes described in the manuscript, including the supporting information. An explanatory phrase was added in the M&M section.

The recent annotation of the Cel epic-1-3 genes was cited in the discussion (lines 522-524). Nevertheless, we did not adopt the numbering 1 – 3, since there is no scientific evidence given for the numbering. It seems to be random. We used x, y, and z instead for different epicuticlins of most species to avoid confusion. Asu-EPIC-1 to 3 have unique protein properties, which allow the numbering.

Asu-epic-1 and 2 are, for example, very different from Cel-epic-1 and 2 and cannot be related directly. NCBI Blast with Asu-epic-1 as a query is not able to detect Cel-epic-1.

---

## [Editor Report · Decision Letter 2]

6 Sep 2022

Identification and characterization of epicuticular proteins of nematodes sharing motifs with cuticular proteins of arthropods

PONE-D-22-12806R2

Dear Dr. BETSCHART,

We’re pleased to inform you that your manuscript has been judged scientifically suitable for publication and will be formally accepted for publication once it meets all outstanding technical requirements.

Kind regards,

Denis Dupuy, Ph.D.

Academic Editor

PLOS ONE
---

## [Editor Report · Acceptance letter]

11 Oct 2022

PONE-D-22-12806R2 

Identification and characterization of epicuticular proteins of nematodes sharing motifs with cuticular proteins of arthropods 

Dear Dr. Betschart:

I'm pleased to inform you that your manuscript has been deemed suitable for publication in PLOS ONE. Congratulations! Your manuscript is now with our production department. 

Kind regards, 

on behalf of

Dr. Denis Dupuy 

Academic Editor

PLOS ONE